# The Power of Genomic in situ Hybridization (GISH) in Interspecific Breeding of Bulb Onion (*Allium cepa* L.) Resistant to Downy Mildew (*Peronospora destructor* [Berk.] Casp.)

**DOI:** 10.3390/plants8020036

**Published:** 2019-02-04

**Authors:** Ludmila Khrustaleva, Majd Mardini, Natalia Kudryavtseva, Rada Alizhanova, Dmitry Romanov, Pavel Sokolov, Grigory Monakhos

**Affiliations:** Center of Molecular Biotechnology, Russian State Agrarian University-Moscow Timiryazev Agricultural Academy, 49, Timiryazevskaya Str., 127550 Moscow, Russian Federation; mr.majdmardini@gmail.com (M.M.); natalia.kudryavtseva92@gmail.com (N.K.); rada.aliz@mail.ru (R.A.); akabos1987@gmail.com (D.R.); pav2395147@yandex.ru (P.S.); breedst@mail.ru (G.M.)

**Keywords:** genomic in situ hybridization (GISH), *Allium cepa*, *A. roylei*, downy mildew resistance, lethal factor, DMR1 marker

## Abstract

We exploited the advantages of genomic in situ hybridization (GISH) to monitor the introgression process at the chromosome level using a simple and robust molecular marker in the interspecific breeding of bulb onion (*Allium cepa* L.) that is resistant to downy mildew. Downy mildew (*Peronospora destructor* [Berk.] Casp.) is the most destructive fungal disease for bulb onions. With the application of genomic in situ hybridization (GISH) and previously developed DMR1 marker, homozygous introgression lines that are resistant to downy mildew were successfully produced in a rather short breeding time. Considering that the bulb onion is a biennial plant, it took seven years from the F_1_ hybrid production to the creation of S_2_BC_2_ homozygous lines that are resistant to downy mildew. Using GISH, it was shown that three progeny plants of S_2_BC_2_ possessed an *A. roylei* homozygous fragment in the distal region of the long arm of chromosomes 3 in an *A. cepa* genetic background. Previously, it was hypothesized that a lethal gene(s) was linked to the downy mildew resistance gene. With the molecular cytogenetic approach, we physically mapped more precisely the lethal gene(s) using the homozygous introgression lines that differed in the size of the *A. roylei* fragments on chromosome 3.

## 1. Introduction

Genomic in situ hybridization (GISH) is a modification of FISH (fluorescence in situ hybridization), which was introduced into molecular cytogenetics almost three decades ago [1]. Since then, GISH has become one of the most powerful tools to analyze natural polyploids, hybrid plants and their backcross progenies for alien gene introgressions, genomic composition, intergenomic rearrangements and the integration of chromosome and recombination maps [2,3,4,5,6]. The great advance in GISH lies in its ability to distinguish between parental genomes in interspecific plant hybrids with no sequence knowledge required. GISH was successfully used to characterize the chromosomal constitutions of tomato somatic hybrids [2]; to prove the allodiploid nature of *Allium wakegi* Araki [7]; to achieve Allium introgression breeding [8,9]; to search for chromosome pairings in allotetraploid hybrids of *Festuca pratensis* × *Lolium perenne* [10]; and to study the origin and evolution of allopolyploids [11]. Despite the simplicity of GISH prerequisites, the method is not used often in actual breeding processes because it is slow and labor intensive. Molecular markers are an effective method for tracking valuable genes, which allows for the acceleration of the breeding [12]. However, molecular markers provide information only across chromosomal regions in linkage disequilibrium with the target trait. The aim of interspecific breeding is to transfer only a gene of interest from wild related species to cultivated crops without a wild background. A huge number of molecular markers are required in order to analyze the whole chromosomes of parental complements while GISH distinguishes the parental genomes and thus allows the introgression of alien material to be monitored at the chromosome level.

Downy mildew (*Peronospora destructor* [Berk.] Casp.) is the most destructive fungal disease for bulb onions (*Allium cepa* L.) during cultivation and storage. The disease causes a significant decrease in bulb yield and a complete loss in seed production under cool moist weather. Up to twelve fungicidal treatments are needed to control the disease spread and these treatments have a negative impact on the environment and human health [13,14]. Bulb onions that are resistant to downy mildew can solve these problems. However, the gene pool of *A. cepa* is rather depleted because of its long history of more than 5000 years of human cultivation [15]. Wild species can be used as donors of economically desirable traits in bulb onion breeding. Thus, the search for donors of resistance genes to downy mildew among closely related wild species was executed. Kofoet and Zinkernagel [16] conducted a highly reproducible screening experiment among 250 cultivars and 24 species of the Allium family, which found complete resistance to downy mildew in *Allium roylei* Stearn. Kofoet et al. [17] reported that the resistance to downy mildew in *A. roylei* could be explained by the presence of a single dominant locus, which was named *Pd_1_*. Random amplified polymorphic DNA (RAPD) marker that was located 2.6 cM from the *Pd_1_* locus was identified [18]. The RAPD marker was converted into a sequence characterized amplified region (SCAR) marker [19,20]. The SCAR marker was mapped on the molecular linkage map based on the *A. cepa* × *A. roylei* cross and the linkage group was assigned to chromosome 3 [20,21]. However, the SCAR marker lost its discriminating power in advanced backcrossed populations [14]. Using GISH, it was shown that *Pd_1_* gene is located in the distal region of the long arm of chromosome 3 and the existence of a recessive lethal gene(s) linked to the resistance gene was verified [14]. A recombinant containing a crossover between the lethal gene(s) and the downy mildew resistance gene was identified [14]. Four AFLP markers closely linked to the resistance gene were identified using homozygous introgression lines [14]. However, the attempt of other breeders to use the AFLP markers to track the introgression of the *Pd_1_* gene failed because they could not reproduce the expected size of the AFLP markers for unknown reasons [22]. The journey of tracking the downy mildew resistance gene from *A. roylei* culminated with the development of a simple robust PCR marker DMR1 based on a high-resolution linkage map [22,23].

In this paper, we describe the advantages of GISH, which is able to act more precisely during the selection process and bridge the gap within the incomplete estimations of alien genetic material that are usually driven by markers. Using GISH and previously developed DMR1 marker that is closely linked to *Pd_1_* gene [22] the lines bearing the homozygous *Pd_1_* gene resistant to downy mildew in the *A. cepa* background were selected. It took 7 years from the F_1_ hybrid production to the creation of S_2_BC_2_ homozygous lines that are resistant to downy mildew. Moreover, GISH visualization of recombination sites on chromosome 3 in homozygous resistant lines allowed us to define a more precise location of the recessive lethal gene(s). 

## 2. Results

### 2.1. DMR1 marker-assisted screening

The robust PCR marker DMR1 linked to the *Pd_1_* downy resistance gene was developed by Kim et al. [22] using downy mildew resistant cultivar ‘Santero’, the linkage maps developed by Duangjit et al. [23] and transcriptome database that has been produced by the RNA-seq analysis [24]. The amplification of a large number of susceptible onions gave a 438-bp product while the resistant cultivar Santero and *A. roylei* produced a 505 PCR product [22]. 

We successfully used the DMR1 marker to analyze F_1_, BC_1_, BC_2_, S_1_BC_2_ and S_2_BC_2_ populations. The non-resistant inbreed lines of *A. cepa* produced a 438-bp PCR product (Figure 1, lines 1,2,3). *A. roylei* that is resistant to downy mildew produced a 505-bp PCR product (Figure 1, lines 4). As expected, F_1_ hybrids between *A. cepa* var. ’Exhibition MS’ and *A. roylei* produced PCR products with both sizes (Figure 1, lines 5). One highly fertile resistant BC_2_ plant was selected for further crossings (Figure 1, lines 6). PCR analysis of 47 S_1_BC_2_ plants revealed one plant that was heterozygous for the *Pd_1_* gene (Figure 1, lines 7) and one plant that was homozygous for the *Pd_1_* gene (Figure 1, lines 8) among the 47 S_1_BC_2_ offspring. The homozygous plant accession S_1_BC_2_-8 was selfed and a PCR analysis of S_2_BC_2_-8 offspring confirmed that all plants were homozygous for the *Pd_1_* gene. The commercial resistant cultivar ‘Santero’ was used as a positive control in PCR analysis (Figure 1, lines 9).

### 2.2. GISH Analysis

Previously, it was established that the *Pd_1_* gene is located in the distal region of the long arm of chromosome 3, which is linked to a recessive lethal gene(s) [14].

#### 2.2.1. S_1_BC_2_

GISH analysis of the *Pd_1_* homozygous accession S_1_BC_2_-8 showed the presence of three recombinant chromosomes in the *A. cepa* genetic background (Figure 2A). Chromosome 3 contained the *A. roylei* fragments in the distal regions on the long arms of both homologs. The *A. roylei* fragments differed in size and contained a large and a small *A. roylei* fragment (Figure 2A’). The size of the large fragment averaged 42.8 ± 2.25% of the length of the long arm and the size of the small fragment averaged 22.3 ± 1.73% (Table 1). The third recombinant chromosome was chromosome 4 (centromeric index: 39.3 ± 2.1 and the relative chromosome length −12.6 ± 0.7). The short arm of chromosome 4, including the centromere and the proximal part of the long arm, originated from *A. roylei* (Figure 2A,A’). The relative position of recombination site was 54.0 ± 1.73% (i.e., the size of *A. cepa* fragment was 54.0 ± 1.73% of the long arm).

#### 2.2.2. S_2_BC_2_

In order to obtain the genotype that carried only the homozygous fragments with *Pd_1_* gene in the *A. cepa* genetic background, a second self-cross was carried out. GISH analysis of S_2_BC_2_-8 offspring allowed us to select three plants (8-33, 8-53 and 8-59) that possessed the *A. roylei* homozygous fragments at the distal region of the long arm of chromosome 3 only in the *A. cepa* genetic background. In offspring 8-33 (Figure 2B,B’), the size of the small fragment was the same as in the parental genotype S_1_BC_2_-8 but the size of the large fragment was 39.7 ± 1.56%, which was significantly different to that of the parental genotype (Mann–Whitney U-test; n1 = n2 = 23; U = 4.5, *P* = 0.01). In offspring plants 8-53 and 8-59 (Figure 2C,C’) the size of both fragments were the same as in the parental genotype (Table 1; Mann–Whitney U-test; n1 = n2 = 6; U = 7, *P* = 0.12).

#### 2.2.3. Monitoring of Crossing over Events between the Recombinant and A. cepa Chromosomes

GISH analysis revealed eight genotypes among 11 S_2_BC_2_ offspring that possessed recombinant chromosome 4 along with a pair of recombinant chromosomes 3. We used these unique genotypes to analyze the location of recombination sites along chromosomes 3 and 4. In offspring 8-5, the position of the recombination site of the large fragment (38.6 ± 1.38%) on chromosome 3 was significantly different from the position in the parental genotype S_2_BC_2_-8, which indicates that a crossover event took place. The small fragment on chromosome 3 (20.3 ± 0.69%) and recombinant chromosome 4 were the same as the parental genotype. Remarkably, in accession 8-20, the size of the large fragment on chromosome 3 was smaller (24.2 ± 1.73%) and nearly was the same size as the small fragment on the homologous chromosome 3 (20.9 ± 1.21%). The recombinant chromosome 4 of offspring 8-20 was crossed with the *A. cepa* homologous chromosome 4 (Figure 2D,D’), which resulted in the recombination site being located on the short arm (44.5 ±2.76%). In offspring 8-32, the size of the small fragment on chromosome 3 increased significantly up to 31.2 ± 1.00% compared to the parental genotype, which suggests a cross-over event (Figure 2E,E’). The large fragment on chromosome 3 and recombinant chromosome 4 of offspring 8-32 were of the parental type. The offspring 8-37 were of the parental types for all three recombinant chromosomes. In offspring 8-38, both *A. roylei* fragments on homologous chromosome 3 have smaller sizes, which reduced the differences between the large and small fragments (Table 1). In contrast to all analyzed accession, both homologous chromosomes 4 were found to be recombinant: one homolog was parental type and another homolog possessed the recombination site on the short arm (46.2 ± 2.60%) along with recombination on the long arm that was of the parental type (Figure 2F,F’). The offspring 8-40 and 8-45 had the same position of recombination sites on the pair of chromosome 3 (Table 1) and a parental-type recombinant chromosome 4 (Figure 2G,G’). In offspring 8-56, the position of a recombination site on homologous chromosome 3 was the same as that of 8-40 and 8-45 while one homologous chromosome 4 was the non-recombinant chromosome of *A. cepa* and another recombinant chromosome 4 was the same as that of 8-38 (Figure 2H,H’). GISH with a commercial cultivar ‘Santero’ confirmed the heterozygous nature of the downy mildew resistance locus and the only fragment of 17.8 ± 1.90% was found on the single chromosome 3 (Figure 2I,I‘). 

GISH visualization of the recombination sites on chromosomes allowed us to reconstruct the crossing-over events of the S_1_BC_2_-8 plant. Six types of recombinant chromosomes 3 and three types of recombinant chromosomes 4 were in viable gametes that formed zygotes and produced the S_2_BC_2_-8 offspring (Figure 3). In two S_2_BC_2_ offspring, the position of the recombination sites in the homologous chromosome 3 with a large fragment was significantly different from that in the parental genome, which resulted in the shortening of the *A. roylei* fragments. The position of the recombination sites in the homologous chromosome 3 with a small fragment also was significantly different in two S_2_BC_2_ offspring compared to that in the parental genome, which resulted in the extending of the *A. roylei* fragments. Thus, GISH analysis indicates at least two recombination events between recombinant chromosomes 3. Gametes bearing the recombinant chromosomes with the *A. roylei* fragments, which were modified in length, were fused in a particular combination in a zygote. GISH revealed one event of recombination between the recombinant chromosome 4 and the non-recombinant *A. cepa* homolog (Figure 3).

## 3. Discussion

In this study, the simplicity of GISH technique in distinguishing parental genomes at the chromosome level in interspecific hybrids together with the previously developed DMR1 marker that is closely linked to *Pd_1_* gene [22] were combined to obtain breeding forms of onion resistant to downy mildew. The exclusivity of this work is that both the GISH technique and the molecular marker were used to follow the breeding process and to conduct targeted selection. In a rather short period of time (7 years), we were able to obtain homozygous introgression lines that were resistant to downy mildew. GISH revealed three offspring of S_2_BC_2_, which possessed the *A. roylei* homozygous fragments at the distal region of the long arm of homologous chromosomes 3 in a complete *A. cepa* genetic background. 

Previously, a recessive lethal gene(s) was found to be linked to the resistance gene [14]. Authors hypothesized that the *A. roylei* fragment of 42.8 ± 1.09% contains the recessive lethal factor that is proximally located to the downy mildew resistant gene while the homozygous fragment of 17.9 ± 0.78% does not bear the lethal factor. Using GISH, we identified the large fragment of 42.8 ± 2.25% of *A. roylei* at the distal region of the long arm of chromosome 3 and the small fragment of 22.3 ± 1.73% in the S_1_BC_2_ genotype. It means that this genotype most likely contains the heterozygous lethal factor. In order to achieve more precise mapping of the lethal factor locus on physical chromosome 3, we measured the position of recombination sites and analyzed the combination of the large and small fragments in the homozygous introgression genotypes of S_2_BC_2_. The largest *A. roylei* homozygous region was found in accession 8-32, which had the small *A. roylei* fragment of 31.2 ± 1.00% and the large fragment of 41.5 ± 1.20% (Table 1). Thus, the *A. roylei* genetic material that was located below 31.2 ± 1.00% were present in both recombinant chromosomes, which showed that it was homozygous (Figure 4). We suggest that the lethal locus located in the *A. roylei* region stretched from 31.2 ± 1.00% to 42.8 ± 2.25% of the long arm of chromosome 3. Given the fact that the smallest *A. roylei* fragment was found in Santero, we conclude that the resistance gene is located below 17.8 ± 1.90% in the distal region (Figure 4). Kim et al. [22] developed two recombinant selection markers DMR2 and DMR3 based on the Santero analysis and linkage map of chromosome 3 [23]. Thus, further analysis of genotypes developed in this study using molecular markers positioned above 163.0 cM [23] will shed light on the nature of the lethal factor. 

A restriction of gene transfer from a wild relative to a cultivated plant associated with the lethal factor was reported for sunflowers in the introgression of downy mildew resistance from *Heliantus tuberosus* [25]. Gametocidal genes (*Gc*) have been reported in different species of Aegilops and belong to the sections Aegilops (*Ae. geniculata* and *Ae. triuncialis*), *Cylindropyrum* (*Ae. caudata* and *Ae. cylindrica*) and Sitopsis (*Ae. longissima*, *Ae. sharonensis* and *Ae. speltoides*) that causes the death of the interspecific hybrid zygotes during early embryonic development [26,27]. In tomatoes, introgression lines were developed to transfer desirable genes from *Solanum pennellii* [28], *S. habrochaites* [29], *S. habrochaites* [30] and *S. lycopersicoides* [31]. In a number of cases, it was difficult or even impossible to obtain homozygous introgression of valuable alien genes, which also might point to the involvement of lethal factors. Removal of such factors will permit successful alien gene introgression.

Recombination took place between *A. cepa* and *A. roylei* within the proximal part of the large introgression fragment. It is known that the chiasmata of *A. cepa* [32] and *A. roylei* [33] are randomly distributed. Chiasma is a cytological manifestation of crossing over. The chromatids of *A. cepa* and *A. roylei* are readily recombined in their interspecific hybrids [8,9]. Thus, the development of molecular markers closely linked to the lethal factor would significantly speed up the selection of a recombinant containing a crossover between the lethal gene(s) and the downy mildew resistance gene. An example of successful breeding was demonstrated on lettuce resistance to the aphid (patent: WO 1997/046080). The aphid resistance was closely linked to an undesirable plant phenotype and this linkage drag could only be removed with the help of molecular markers and large segregating populations.

## 4. Materials and Methods

### 4.1. Plant Materials

*A. cepa* (2n = 2C = 16) inbred lines were from the genetic collection of the N. N. Timofeev Breeding Station. *A. roylei* Stearn (2n = 2C = 16) was obtained from the Centre for Genetic Resources in the Netherlands and was used as the pollen parent to be crossed over with the cytoplasmic-male-sterile (T) line ‘Exhibition’ in order to produce the interspecific F_1_ hybrids. The following first and second backcrosses were conducted using the inbred lines ‘Hiberna’ and ‘Valensia’, respectively, as the male parents and selfing took place at the end of the breeding scheme (Figure 5). A fertile BC_2_ plant resistant to downy mildew was selected under the conditions of natural infection in the greenhouse experiment using a susceptible variety that served as the spreader plants to obtain sufficient disease pressure. Furthermore, we used PCR analysis with DMR1 marker to confirm that the selected fertile resistant BC_2_ plant possessed the heterozygous *Pd_1_* gene. This BC_2_ plant was self-pollinated. PCR analysis of S_1_BC_2_ offspring showed only two genotypes (plants 7 and 8) possessing the resistance gene among the 47 S_1_BC_2_. These plants 7 and 8 was analyzed with GISH. A second self-cross of the plant 8 possessing homozygous *Pd_1_* gene was conducted. The 30 seeds of the S_2_BC_2_-8 offspring were sown in greenhouse in 2018 for GISH screening. The seed germination was 85.0%.

We used the ‘speed breeding’ protocol to produce F_1_, BC_1_ and BC_2_ generations. To reduce the generation time, we used prolonged photoperiod and vernalization to obtain mature seeds in one year instead of two years. F_1_, BC_1_ and BC_2_ seeds were sown in pots and were grown for a 16-h photoperiod (REFLUX lamp 400 watts; light intensity: 8000 lx) at a temperature of 22 °C during the photoperiod and 14 °C during the 8-hour dark period. This was maintained at 80% humidity for about five months (August–December). There was plant vernalization for two months (January–February) at 6 °C with the 16-h photoperiod. In March, the plants were transferred to an unheated greenhouse under natural light at a temperature of 10–14 °C during the day and 6–8 °C at night. The following stages occurred in certain months as described: in April—bolting and in May—flowering and crossing; and in July—seed harvesting. The self-crossing S_1_BC_2_ and S_2_BC_2_ were produced using the common method for two years (seed-set-seed) as follows: seeds were sown in small pots that were Plantek 144 (pH of soil 5.5–5.6; N: 130 mg/L; P: 205 mg/l; K: 280mg/L) in the middle of April. The seedlings were transplanted in the field in the middle of May. The onion sets were harvested at the beginning of September and in February, they were planted in pots in green house and were grown at 4–8 °C. From the beginning of April, the temperature was increased up to 22 °C during the day and 14 °C at night.

### 4.2. PCR Analysis with DMR1

Genomic DNA was isolated from green leaves by the CTAB-method [34]. PCR products were obtained using a pair of primers DMR1-F1 and DMR1-R1 [22]. The sequence of the used primer is: 5’TGAGGCTCAAGTTGACATGC3’, forward; and 5’TTCGTAGCAGCATCAAGGTG3’, reverse. PCR was conducted in a 25-µL reaction mixture (1x Taq buffer; 2.5 units Taq-polymerase (Sileks, Moscow, Russia); 2.5 mM MgCl2; 0.2 mM dNTP; 0.2 µM primers; 50 ng DNA). The PCR amplification procedure consisted of the following steps: initialization step at 95 °C for 4 min; 35 cycles at 95 °C for 60 s; 52 °C for 60 s and 72 °C for 60 s; and a final 10 min elongation at 72 °C. The PCR products were visualized in 2% agarose gel with ethidium bromide staining.

### 4.3. Preparation of Mitotic Chromosomes

Mitotic chromosomes were prepared from young root tips using the squash method according to Khrustaleva et al. [8] with a slight modification. Briefly, young root tips were pretreated with an aqueous solution of α-bromnaphtalene (1:1000) overnight at 4 °C, fixed in 3:1 (v/v) ethanol-acetic acid for at least 1 h at RT and rinsed three times in distilled water. This was finally incubated in 10 mM citrate buffer (pH 4.8), containing 0.1% (w/v) cellulase RS, 0.1% (w/v) pectolyase Y23 and 0.1% (w/v) cytohelicase, for 75 min at 37 °C. The macerated root tips were spread by being dissected and squashed in a drop of 45% acetic acid. The permanent slides were prepared using liquid nitrogen.

### 4.4. Genomic in Situ Hybridization (GISH)

Genomic DNA was extracted from 4 g of young leaves using the CTAB method of Rogers and Bendich [34]. To create a probe, the total genomic DNA of *A. roylei* was fragmented with Labsonic^®^ M (Sartorius, Göttingen, Germany) and labeled with Dig-11-dUTP (Roche Diagnostics Gmbh, Mannheim, Germany) according to the manufacturer’s instructions. The genomic DNA of *A. cepa* was used as a block DNA. DNA denaturation and in situ hybridization steps were performed according to Schwarzacher and Heslop–Harrison [35]. The hybridization mix contained 50% (v/v) deionized formamide, 10% (w/v) sodium dextran sulfate, 2xSSC, 0.25% (w/v) SDS, 1.0 ng/µL probe DNA and 30µg/µL block DNA. A 78% stringency washing was applied. The Dig-11-dUTP labeled DNA was detected with anti-Digoxigenin-FITC antibody raised in sheep (Roche Diagnostics Gmbh, Mannheim, Germany) and amplified with anti-sheep-FITC antibody raised in rabbits (Vector Laboratories, Burlingame, California, USA). *A. cepa* chromosomes was counter-stained with DAPI in Vectashield antifade mounting medium at a concentration of 1:20 (Vector Laboratories, California, USA).

### 4.5. Microscopy and Image Analysis and Karyotype Analysis

Slides were examined under a Zeiss Axio Imager microscope (Carl Zeiss MicroImaging, Jena, Germany). The selected images were captured using a sensitive black-white Axio Cam MRm (Carl Zeiss MicroImaging, Jena, Germany) digital camera. Image processing and thresholding were performed using AxioVision version 4.6 software (Carl Zeiss MicroImaging, Jena, Germany). The DAPI fluorescence signal was represented in the red pseudo color. Final image optimization was performed using Photoshop (Adobe Inc., San Jose, California, USA).

Karyotype analysis and identification of individual chromosome with fluorescent signals were performed according to the bulb onion nomenclature [36] and previously published karyotypes of closely related Allium species [37]. The chromosome measurements and karyotyping were performed by using a freeware computer application DRAWID [38]. The relative position of recombination site was calculated as a ratio between the distance from the distal end of the chromosome arm and recombination site to the length of the arm.

## 5. Conclusions

In conclusion, our study demonstrated the power of GISH in plant interspecific breeding, which allowed us to monitor both the introgression of the alien target gene and unwanted genetic material of wild relatives. Using a combination of GISH and the previously developed DMR1 marker that is closely linked to *Pd_1_* gene [22], the homozygous introgression lines resistant to downy mildew were successfully produced in a rather short time period (7 years). Moreover, the lethal gene(s) linked to *Pd_1_* gene was more precisely mapped on chromosome 3 that will help in developing markers that are closely linked to the lethal gene(s) and will speed introgression breeding of other onions resistant to downy mildew.

## Figures and Tables

**Figure 1 plants-08-00036-f001:**
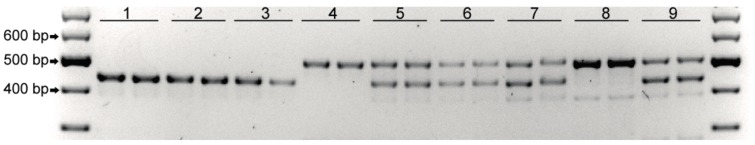
Monitoring the introgression of downy mildew resistance locus from *A. roylei* into *A. cepa* genome using PCR with DMR1 marker [22]. Lines 1: *A. cepa* var. ‘Exhibition MS’; lines 2: *A. cepa* var. ‘Hiberna’, lines 3: *A. cepa* var. ‘Valencia’; lines 4: *A. roylei*; lines 5: F_1_
*A. cepa* var. ‘Exibition MS’ × *A. roylei*; lines 6: BC_2_ [(*A. cepa* var. ‘Exhibition MS’ × *A. roylei*) × *A. cepa* var. ‘Hiberna’ × *A. cepa* var. ‘Valencia’]; lines 7, 8: S_1_BC_2_; lines 9:resistant cultivar ‘Santero’. Flanking gel tracks form a 100-bp DNA ladder (Thermo Scientific).

**Figure 2 plants-08-00036-f002:**
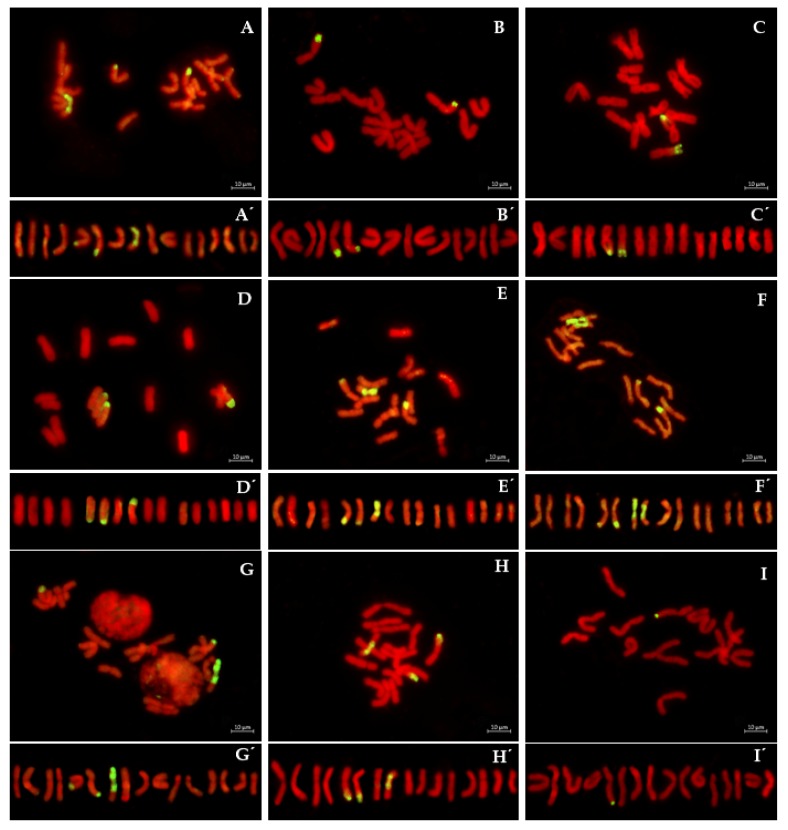
Genomic in situ hybridization (GISH) analysis of S_1_BC_2_ and S_2_BC_2_ introgression homozygous lines resistant to downy mildew. (**A**) S_1_BC_2_-8. S_2_BC_2_-8: (**B**) 8-33, (**C**) 8-53, (**D**) 8-20, (**E**) 8-32, (**F**) 8-38, (**G**) 8-45 and (**H**) 8-56. (**I**) Santero. (**A’–I’**) karyotypes of the corresponding accessions. *A. roylei* genetic material appears in yellow and *A. cepa* genetic material appears in red.

**Figure 3 plants-08-00036-f003:**
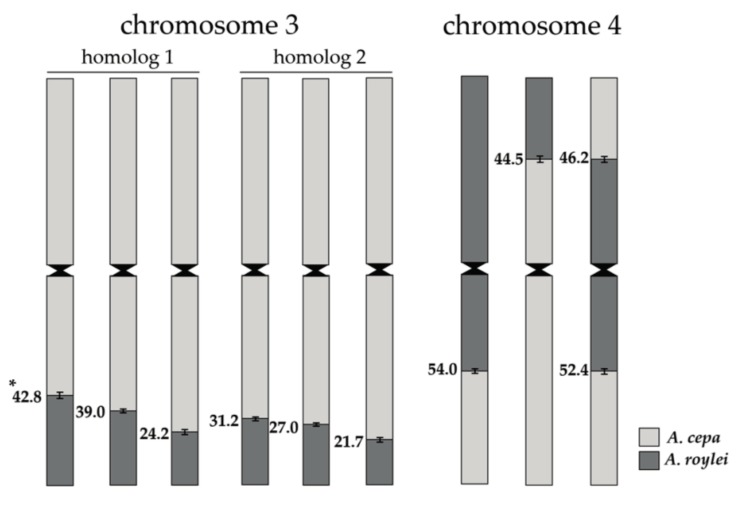
Types of recombinant chromosomes in S_2_BC_2_ offspring. * Position of recombination site is the ratio of the length of the distance between distal end of chromosome arm and recombination site to the total length of the arm (%).

**Figure 4 plants-08-00036-f004:**
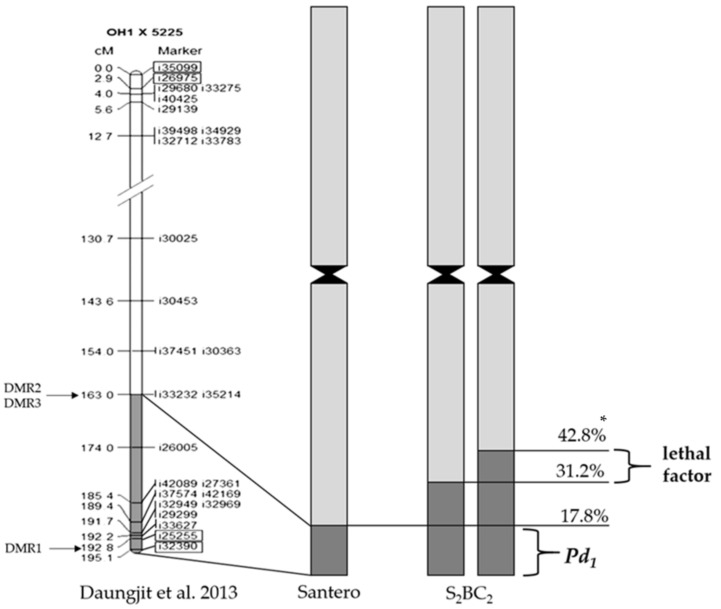
The integrated recombination [23] and physical maps for the S_2_BC_2_ recombinant *A. cepa* × *A. roylei* chromosome 3. DMR1: the marker linked to downy mildew resistance gene; DMR2 and DMR3: the recombinant selection markers, which were produced by Kim et al. [22]. Loci used by Kim et al. [22] in the identification of the position of the resistance gene are enclosed in rectangular boxes. * Position of recombination site is the ratio of the length of the distance between distal end of chromosome arm and recombination site to the total length of the arm (%).

**Figure 5 plants-08-00036-f005:**
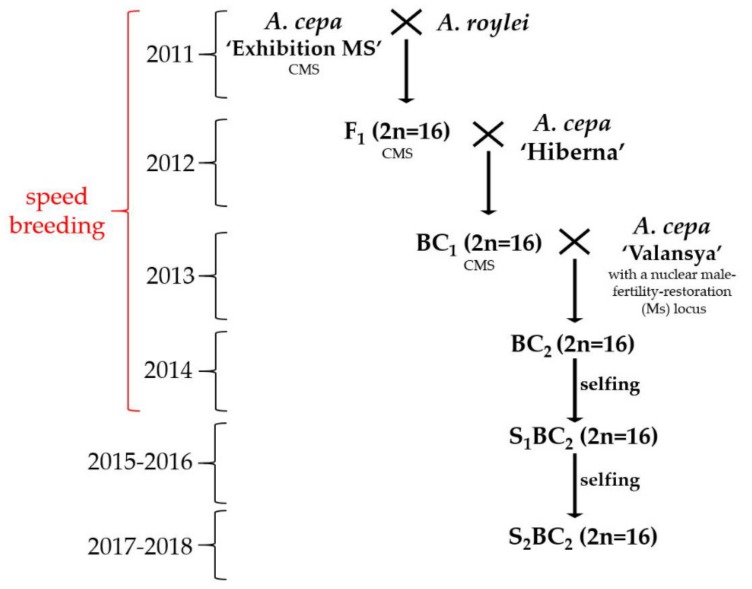
The scheme of interspecific breeding of bulb onion resistant to downy mildew. ‘Speed breeding’: using prolonged photoperiod and vernalization for obtain of mature seeds in one year. CMS: cytoplasmic mail sterility.

**Table 1 plants-08-00036-t001:** The position of recombination sites between *A. cepa* and *A. roylei* within recombinant chromosome 3 and 4 in S_1_BC_2_ and S_2_BC_2_ genotypes.

Accession		Position of recombination site*, %			
		Chromosome 3		Chromosome 4	
		Homologous 1 (large fragment) Mean ± SD	Homologous 2 (small fragment) Mean ± SD	Homologous 1 Mean ± SD	Homologous 2 Mean ± SD
**S1BC2-8 (parental form)**		42.8 ± 2.25	22.3 ± 1.73	*A. cepa***	54.0L ± 1.73
**S2BC2**	8-5	38.6 ± 1.38	20.3 ± 0.69	*A. cepa*	54.0L ± 1.73
	8-20	24.2 ± 1.73	20.9 ± 1.21	*A. cepa*	44.5S ± 2.76
	8-32	41.5 ± 1.20	31.2 ± 1.00	*A. cepa*	54.0L ± 2.00
	8-33	39.7 ± 1.56	22.5 ± 4.02	*A. cepa*	*A. cepa*
	8-37	42.4 ± 1.56	22.5 ± 0.69	*A. cepa*	54.0L ± 1.73
	8-38	39.1 ± 1.00	27.0 ± 1.40	54.0L ± 2.00	46.2S ± 2.60 52.4L ± 2.40
	8-40	38.2 ± 2.25	24.7 ± 1.56	*A. cepa*	54.0L ± 1.73
	8-45	39.4 ± 1.60	24.1 ± 1.40	*A. cepa*	54.0L ± 2.00
	8-53	42.3 ± 1.56	24.1 ± 1.65	*A. cepa*	*A. cepa*
	8-56	39.0 ±1.00	24.2 ± 2.00	*A. cepa*	51.0S ± 2.00 54.6L ± 2.00
	8-59	41.8 ± 1.56	23.0 ± 1.38	*A. cepa*	*A. cepa*
**Santero**		*A. cepa*	17.8 ± 1.90	*A. cepa*	*A. cepa*

* Position of recombination site is the ratio of the length of the distance between distal end of chromosome arm and recombination site to the total length of the arm. ** Non-recombinant *A. cepa* chromosome. ^L^ Long arm. ^S^ Short arm.

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
