# Peer review of "The Power of Genomic in situ Hybridization (GISH) in Interspecific Breeding of Bulb Onion (Allium cepa L.) Resistant to Downy Mildew (Peronospora destructor [Berk.] Casp.)"

_plants, 2019, doi:10.3390/plants8020036_

Reviewer 1 Report

The paper is worth publishing, but need extensive editing of English!!!

Author Response

We are grateful for evaluating our research and appreciate the time you spent reading our manuscript shortly before Christmas and New Year! 

We made changes to the introduction. We have made efforts to improve our English.

Reviewer 2 Report

The paper of Khrustaleva et al. is an interesting application of GISH to monitor the introgression process at the chromosome level in interspecific breeding of Allium cepa. The results demonstrated the power of this molecular cytogenetic technique for this specific study since revealed the location of the fragments of A. royle and the precise positon of the lethal genes. The paper is well written, the methods and the results are clearly described. On my opinion this paper is suitable for publication in PLANTS journal.  My only suggestion is to make shorter the Introduction in the first part.

Author Response

We are grateful for evaluating our research and appreciate the time you spent reading our manuscript shortly before Christmas and New Year! 

We made changes to the introduction.

Reviewer 3 Report

I downloaded the manuscript and with track changes and add comments I improved the text and asked my questions. Especially I missed the arguments why GISH really has more power than other techniques.

Author Response 

We are grateful for evaluating our research and appreciate the time you spent reading our manuscript shortly before Christmas and New Year! Thank you very much for your interesting questions and improving English language in the text.

Our answers to your questions are submitted in the attached file. 

Round  2

Reviewer 1 Report

The review is provided in separate file

Author Response

We are grateful for such a detailed analysis of our research and your comments, which improved our manuscript. Thank you so much for correcting the English language. Answers to your comments are in the attached file.

Your comment: The authors were able to discriminate the positions of the recombination events in the resolution of percentages (or order below this!!!) of the length of the arm. See the figure (for example 2D): I have hard time to find the centromeres and the authors were able to discriminate the recombination in x.x%!!!

Answer: The calculation of the length of the arm and determining the position of the centromere we carried out on DAPI images, and not on the merged images as is customary in molecular cytogenetics.  The size of the alien   fragment is determined on the images obtained on the filters of a specific fluorochrome, which is used in the probe labeling or detection of stable hybridization sites. We used FITC images. As you requested, we send five metaphases illustrating the measurement procedure (see attached file). I hope this will convince you of the reliability of our results.

Your comment: With unbelievable SD! How many metaphases were used for each accession? It is not mentioned in Methods

Answer:  Measurement of chromosomes was carried out in 5-10 metaphases for each accession. We added this sentence in Methods.

We looked at our Excel data. You're right. This is an annoying typo. This is not standard deviation (SD). This is Standard Error of Mean (SEM) which is always less than SD.

We analyzed the literature and consulted with our statistical experts. We concluded that in our case should be used SD (Barde, M. P., & Barde, P. J. (2012). What to use to express the variability of data: Standard deviation or standard error of mean? Perspectives in clinical research3(3), 113-60).  Thus, we changed Table1.

Your comment Is it really complete A. cepa background? I would assume that the part of the substitution is missing…

Answer:  Even if there was a high-dense map covering all the chromosomes of A. roylei and A. cepa, there would be doubt that all the genetic material of A. roylei was detected in the hybrid. To be 100% sure you need to make full-genome sequencing of A. roylei, three lines of A. cepa and hybrids.

Round  3

Reviewer 1 Report

I think that manuscript was edited enough to be accepted.

Author Response

Thank you